# Enhanced Adsorption of Carbon Dioxide from Simulated Biogas on PEI/MEA-Functionalized Silica

**DOI:** 10.3390/ijerph17041452

**Published:** 2020-02-24

**Authors:** Yankun Sun, Wanzhen Liu, Xinzhong Wang, Haiyan Yang, Jun Liu

**Affiliations:** 1College of Resource and Environment, Northeast Agricultural University, Harbin 150030, China; sunyankun@neau.edu.cn (Y.S.); liuwanzhen9023@163.com (W.L.); wangxinzhong1902@163.com (X.W.); 2College of Arts and Sciences, Northeast Agricultural University, Harbin 150030, China; 3School of Mechanical Engineering, Harbin Vocational and Technical College, Harbin 150030, China; k___h@163.com

**Keywords:** silica, solid amine, biogas

## Abstract

A series of efficient adsorbents were prepared by a wet-impregnation method for CO_2_ separation from simulated biogas. A type of commercially available silica, named as FNG-II silica (FS), was selected as supports. FS was modified with a mixture of polyethyleneimine (PEI) and ethanolamine (MEA) to improve the initial CO_2_ adsorption capacity and thermal stability of the adsorbents. The influence of different adsorbents on CO_2_ adsorption performance was investigated by breakthrough experiments. Scanning electron microscopy (SEM), fourier transform infrared spectroscopy (FTIR), and N_2_ adsorption–desorption isotherm were used to characterize the silica before and after impregnating amine. Additionally, the thermal stability of adsorbents was measured by differential thermal analysis (TDA). Silica impregnated with mixtures of MEA and PEI showed increased CO_2_ adsorption performance and high thermal stability compared with those obtained from silica impregnated solely with MEA or PEI. With a simulated biogas flow rate of 100 mL/min at 0.2 MPa and 25 °C, FS-10%MEA-10%PEI exhibited a CO_2_ adsorption capacity of ca. 64.68 mg/g which increased by 81 % in comparison to FS-20%PEI. The thermal stability of FS-10%MEA-10%PEI was evidently higher than that of FS-20%MEA, and a further improvement of thermal stability was achieved with the increasing value of PEI/MEA weight ratio. It was showed that MEA was able to impose a synergistic effect on the dispersion of PEI in the support, reduce the CO_2_ diffusion resistance and thus increase CO_2_ adsorption performance. Additionally, if the total percentage of amine was the same, FS impregnated by different ratios of PEI to MEA did not exhibit an obvious difference in CO_2_ adsorption performance. FS-15%PEI-5%MEA could be regenerated under mild conditions without obvious loss of CO_2_ adsorption activity.

## 1. Introduction

Carbon dioxide (CO_2_) released from fossil fuel combustion is considered to be the main cause of the greenhouse effect. Therefore, the use of renewable clean energy to substitute fossil energy has become one of the main approaches in CO_2_ emission reduction policies [1,2,3,4,5]. As a clean energy source, biogas consists of 40–75 vol% CH_4_, 25–60 vol% CO_2_, and other compounds (H_2_O, H_2_S, N_2_, O_2_, NH_3_, and siloxanes) [6,7,8]. The presence of a large amount of CO_2_ will reduce the calorific value of biogas [8] and increase the cost of transportation and storage, thereby limiting the application of biogas as a renewable energy source. The removal of CO_2_ improves the fuel rating [9] and enhances the cost-effectiveness of the biogas. In addition, the separated CO_2_ can be used as an alternative raw material for producing high value chemicals, which is also advantageous for delaying global warming. Therefore, it is necessary to develop an economical and effective method for biogas purification.

The most widely used method for separating CO_2_ from biogas is liquid amine absorption [10]. However, this method faces limitations during actual application due to slow regeneration and high cost [11]. Compared to traditional biogas purification technologies, the solid adsorption method has advantages such as low energy consumption and little equipment corrosion [12,13]. Common solid adsorbents include zeolite [14,15], activated carbon [16,17], and metal organic framework (MOF) [11,18]. They have decent CO_2_ selectivity, but they exhibit limited adsorption performance in the presence of water. To overcome these shortcomings, solid amine adsorbents have been prepared by researchers by either impregnating organic amine molecules on porous supports or chemically grafting amino groups onto support surface. Solid adsorption method combined both chemical and physical absorption [19] and improved CO_2_ adsorption selectivity as well as water tolerance [20,21]. Furthermore, this method enables the construction of smaller process equipment, further reducing the cost of CO_2_ separation [15]. Therefore, it is considered to be a promising CO_2_ capture and separation technology. Various research groups have developed various porous materials as supports for amine-functionalized solid adsorbents. Among these, mesoporous silicas are widely used as support materials due to their outstanding textural and surface properties, such as large surface area and pore volume [22,23]. Polyethyleneimine (PEI) has become to be the most typical commercial amine impregnated into mesoporous silica. However, PEI-impregnated silica adsorbents with high amine loading were demonstrated to be unevenly distributed in the porous materials and the agglomerated viscous PEI chains in silica pores hindered CO_2_ diffusion, resulting in the low CO_2_ capacity [24]. An efficient way to increase amine efficiency in amine impregnated adsorbents is to use low molecular weight amine species like tetraethylenepentamine (TEPA) to improve the dispersion of amine and minimize CO_2_ diffusion resistance [12,25,26,27]. However, low molecular weight amine species often had the drawback of leaching out of the supports. Another strategy to increase amine efficiency is to add the additives with hydroxyl groups during adsorbents synthesis step. It was reported that the presence of hydroxyl groups could efficiently improve the dispersion of amine molecule in the support pore surface to increase adsorption capacity. Satyapal et al. [28] developed regenerable PEI based solid amine adsorbents with a polyethylene glycerol (PEG) coating and reported the OH groups in PEG exhibited the positive effects on CO_2_ adsorption. To further improve the performance of the adsorbent, some researchers have added amine species with hydroxyl groups to PEI in order to modify porous supports and have achieved relatively good results. Yue et al. [29] used a mixture of TEPA and diethanolamine (DEA) to modify the mesoporous silica and found that the adsorption capacity of the mixed amine adsorbent (163 mg/g) is significantly larger than those that were modified with TEPA (144 mg/g) or DEA (20.8 mg/g) alone. Xue et al. [30] increased the CO_2_/CH_4_ separation coefficient to more than twice by adding piperazine (PZ) to methyldiethanolamine (MDEA). Fan et al. [24] mixed PEI and DEA to modify silica and enhanced the amine efficiency of PEI/DEA-silica (0.4 mol CO_2_/mol N) more than twice that of PEI-silica (0.17 mol CO_2_/mol N) at 35 °C. Meanwhile, the adsorption capacity of PEI/DEA-silica also increased by 50% compared to the that of PEI-silica.

In this work, we examined the effect of PEI/MEA modified silica and that modified with PEI or ethanolamine (MEA) alone on separating CO_2_ from simulated biogas. A series of solid amine adsorbents were prepared by impregnating a type of popular commercial silica with MEA, PEI and PEI/MEA mixture. The adsorbents were evaluated in terms of CO_2_ adsorption capacity and regeneration performance.

## 2. Experimental

### 2.1. Materials

Commercially available FNG-II silica labeled as FS (pellets, 2~4 mm) was purchased from Qingdao Haiwan Chemical Co., Ltd. (Qingdao, China). Simulated biogas (35.6% CO_2_, 64.4% CH_4_) and Argon (Ar) were purchased from Liming Gas Co., Ltd. (Harbin, China). PEI (99%) with a molecular weight (M_W_) of 1800 and methanol (99.5%) were purchased from Aladdin Industrial Inc. (Shanghai, China). MEA (≥99.0%) were from Tianjin Fuchen Inc. (Tianjin, China). All chemicals were used without additional treatment.

### 2.2. Preparation of Amine-Functionalized Samples

Before used as the support, the FS was dried at 105 °C until no weight loss was observed. A wet impregnation method was used to prepare the PEI-functionalized samples [31,32]. Amine (pure or mixture of two) were dissolved in 60 mL of methanol and then the dry silica was added into the solution. The amine/methanol/silica mixture was continuously soaked for 12 h, followed by the removal of methanol by rotary evaporation from 50 °C to 70 °C gradually under a vacuum of 150 mmHg. The ratio mass/volume of adsorbents reached ca. 480 g/L. The prepared samples were labeled as FS-X% AMINE, where X% represented the weight percentage of the amine in the samples and AMINE is referred to the name of the used amine.

### 2.3. Characterization of the Adsorbents

Before the following determination, the silica was dried while the amine-functionalized silica was measured without additional pretreatment. Nitrogen adsorption and desorption isotherms were performed at 77 K on a surface area and pore size analyzer apparatus (NOVA 2000e, Quantachrome, Boynton Beach, FL, USA). Before each measurement, the sample was degassed at 453 K for 3 h under a vacuum. The specific surface area was calculated from the isotherm data using the Brunauer–Emmett–Teller (BET) and the total pore volume was determined as the volume of liquid nitrogen adsorbed at a relative pressure of 0.97. Pore diameters were evaluated from the desorption branch of the isotherm based on the Barrett–Joyner–Halenda (BJH) model. The SEM images were obtained using a SU8010 microscope (Hitachi, Tokyo, Japan) equipped with a field emission gun. The acceleration voltage was set to 5 kV. The samples were stuck on the observation platform and sprayed with gold vapor under high vacuum for about 20 s. The FTIR spectra were recorded on an ALPHA-T spectrometer (Bruker, Karlsruhe, Germany) using the KBr compression method. The silica was treated at 378 K for 10 h prior to Infrared spectra (IR) analysis. Thermogravimetric (TG) curves were obtained using a STA449F5 thermal analysis instrument (Netzsch, Selb, Germany) in an N_2_ atmosphere.

### 2.4. Adsorption Experiments

Before breakthrough experiments, each amine-functionalized sample was dried to constant weight under 120 °C and the protection of ultrapure Ar to eliminate volatile components. Breakthrough experiments were carried out with a self-assembled fixed bed system (illustrated in Figure 1), which has been reported in our previous work [33]. The testing adsorbent was packed in the adsorption column which was a stainless steel tube with inner diameter of 11 mm and length of 250 mm. In order to desorb the volatiles, the adsorbent was pretreated in advance by heating at 120 °C with an ultrapure Ar stream at a flow rate of 100 mL/min for 30 min. Each adsorption measurement was performed at 0.2 MPa and 25 °C. Prior to adsorption, Ar was introduced into the adsorption column to clear the air in the measurement system and the pressure was increased to 0.2 MPa. After that, a simulated biogas (64.4 vol% CH_4_, 35.6 vol% CO_2_) stream at a total flow rate of 100 mL/min was introduced and passed through the adsorbent. The outlet concentration was determined by gas chromatography (GC), until the outlet concentration reached the initial feed concentration, which indicated the adsorbent had been saturated by CO_2_. There was no appreciable pressure drop across the adsorption column was observed.

The adsorption capacity of CO_2_ on an adsorbent was calculated as: (1)$$Q=\frac{M\times F\times \int_{0}^{t}\left(C_{0}-C\right)dt}{W_{a}\left(V_{a}\right)}\times \frac{T_{0}}{T}\times \frac{1}{V_{m}}$$
where Q is CO_2_ adsorption capacity (mg/g or mg/cm^3^); t is the sorption time (min); F is the flow rate (mL/min); W_a_ is the weight of adsorbent (g); M is the molecular weight (g/mol) of CO_2_; C_0_ and C_t_ are the inlet and outlet concentration (vol%) of CO_2_, respectively; T represents the gas temperature (K); T_0_ is 273 K, and V_m_ is the molar volume (22.4 dm^3^/mol).

## 3. Results and Discussion

### 3.1. Characterization of Materials

The FTIR spectra of FS is depicted in Figure 2. The peak at 3431 cm^−1^ is attributed to the O-H stretching vibrations of the hydrogen-bonded silanol groups and adsorbed water molecules [34]. The peak located at 1628 cm^−1^ was corresponding to adsorbed water [35]. Additionally, the peaks at 800 cm^−1^ and 1107 cm^−1^ are associated with Si-O and Si-O-Si vibrations [36].

The N_2_ adsorption and desorption isotherms of pure FS and amine-functionalized samples are given in Figure 3. FS exhibits type-IV isotherms. At the relative pressures P/P_0_ from 0.45 to 0.95, there is a hysteresis loop which is attributed to the mesopores. The pores distribution curve (BJH) of FS is shown in Figure 4 and its average pore is ca. 8 nm. A shift to larger pores was observed with FS-amine adsorbents (Figure 4). However, the general distribution of the pore size did not change among different FS-amine adsorbents. The presence of these relatively large pores in the adsorbent is beneficial for CO_2_ to access the active amino sites. 

Textural properties of FS before and after loading amine were listed in Table 1. It can be seen that the BET surface and total pore volumes of amine-samples are all lower than those of raw FS, which indicated amine had been introduced into the silica. Compared with raw FS, the pore volume losses of FS-20%PEI, FS-20%MEA, and FS-10%MEA-10%PEI were 0.254076, 0.235357, and 0.227252 cm^3^/g respectively and their BET surface accordingly decreased 132.5689, 108.355, and 126.3962 m^2^/g respectively. FS-20%PEI showed the largest pore volume and surface area losses which strongly suggests that PEI was very unevenly dispersed in FS.

The scanning electron microscope (SEM) morphology of silica before and after impregnating organic amine was shown in Figure 5. It can be seen from SEM photos magnified 10,000 times that FS-20%MEA and FS-10%MEA-10%PEI still retained the original morphology of the support. However, a partial agglomeration of PEI could be observed on the outer surface, which indicates that PEI was not homogeneously dispersed on the surface of FS.

The thermal stabilities of FS before and after loading amine were shown in Figure 6. It could be seen that there was a weight loss before 100 °C for FS (black line) caused by the desorption of adsorbed moisture and no significant weight losses appeared after 100 °C in FS. The overall weight decrease for FS appeared to be 4.3% in the range of 25–800 °C. With regard to FS-20%PEI (green line), the first weight loss stage located before 130 °C arose from the desorption of pre-adsorbed H_2_O and CO_2_. The second stage in the range 230–380 °C was originated from leaching or decomposition of PEI in the FS, which suggested that the PEI-functionalized FS was stable below these temperatures. For FS-20%MEA (blue line), a large weight decrease (ca. 12%) appeared before 130 °C mainly because of leaching of MEA in the FS, suggesting FS-20%MEA had poor thermal stability. FS-10%MEA-10%PEI (red line) exhibited two TG stages and an overall weight decrease of ca. 21% in the range 25–800 °C. Its first weight loss stage appeared before 130 °C and reached ca. 7%, which was mainly originated from leaching of MEA. The second stage in the range 230–380 °C was originated from leaching or decomposition of PEI in the FS. With regard to FS-5%MEA-15%PEI (magenta line), its first weight loss reached to ca. 5%, lower than that of FS-10%MEA-10%PEI, indicating that the thermal stability of FS-5%MEA-15%PEI was higher than that of FS-10%MEA-10%PEI.

### 3.2. Separation of Carbon Dioxide from Simulated Biogas

The CH_4_/CO_2_ gas separation behaviors of the dynamic experiments were illustrated in Figure 7. It can be seen that FS-20%PEI, FS-20%MEA and FS-10%MEA-10%PEI were all able to separate CO_2_/CH_4_ effectively which is indicated by the large differences in the breakthrough times of CO_2_ and CH_4_, but the separation ability of the three samples was quite different. For further studies, adsorption capacity of CO_2_ on the samples are shown in Table 2. It can be observed that the CO_2_ adsorption capacity of FS-20%PEI was much smaller than that of FS-10%MEA-10%PEI and FS-20%MEA, which were mainly because PEI was very unevenly dispersed in FS and thus amino groups were not easily accessible to CO_2_. FS-20%PEI showed larger pore volume and surface area losses than FS-10%MEA-10%PEI and FS-20%MEA did (shown in Table 1) compared with their supports (FS), which just indicated that amine was dispersed more unevenly in FS-20%PEI and thus showed much lower CO_2_ adsorption capacity. The uniformity of the amine dispersion can also be seen in SEM (Figure 5). While FS-10%MEA-10%PEI exhibited a much higher CO_2_ adsorption capacity than FS-20%PEI, this was because MEA mixed with PEI could boost the dispersion of PEI and thus facilitated CO_2_ coming into contact with amino groups. The lower decrease in pore volume and surface area losses of FS-10%MEA-10%PEI can explain this. For FS-20%MEA, its CO_2_ adsorption capacity was slightly higher than that of FS-10%MEA-10%PEI, but it exhibited significant signs of leaching owing to the low volatility of MEA. As for FS-10%MEA-10%PEI, it was more stable with regard to possible leaching problems. Therefore, FS-10%MEA-10%PEI can overcome the weakness of low CO_2_ adsorption capacity on FS-20%PEI and conquer the defect of instability of FS-20%MEA.

The effect of the ratio of PEI to MEA on the CO_2_ adsorption characteristics of PEI/MEA functionalized FS was studied (Figure 8). It is very interesting to note that CO_2_ adsorption capacity of PEI/MEA impregnated FS wasn’t enhanced with the increase of MEA/PEI weight ratio. The similar CO_2_ adsorption capacity was exhibited although different ratios of MEA to PEI were impregnated into FS, which also suggested that for MEA/PEI functionalized FS, it was not MEA itself that played the most significant role on adsorbing CO_2_ but a better dispersion of PEI promoted by MEA. This is probably because MEA facilitated PEI to disperse evenly on the surface of FS and thus allowed CO_2_ to access more easily to the active amino groups. So the PEI/MEA functionalized FS showed much higher CO_2_ adsorption capacity than FS-20%PEI, even when the weight ratio of MEA was as low as 5%.

As TG results showed, improvements of thermal stability would be achieved with the increasing value of PEI/MEA weight ratio, so FS-15%PEI-5%MEA was chosen to evaluate the regeneration performance of FS-PEI/MEA adsorbents. The regeneration performance of the FS-15%PEI-5%MEA adsorbent was evaluated by conducting 6 sequential adsorption–desorption cycles (see Figure 9). The adsorption was performed at 25 °C, 0.2 MPa, as well as a simulated biogas flow rate of 100 mL/min and the regeneration was performed at 130 °C, 0.2 MPa as well as an Ar (99.999%) flow rate of 200 mL/min for 15 min. The breakthrough CO_2_ adsorption capacities did not significantly change on 1st, 2nd, and 3rd cycles, remaining within the narrow ranges of 59.80–64.17 mmol/g (Figure 8). This also indicates that PEI coated on FS did not degrade significantly at 130 °C in the presence of oxygen. The breakthrough CO_2_ adsorption capacities slightly decreased on 4th, 5th and 6th cycles with 55.3–55.54 mmol/g due to accumulation of very small amount of residual CO_2_ in adsorbent and this possibly required longer regeneration time in 4th to 5th cycles for the case of high CO_2_ concentration feed. However, these results showed that the adsorption performance of FS-15%PEI-5%MEA was stable. Therefore, this adsorbent could be regenerated under mild conditions without obvious loss of CO_2_ adsorption activity and so may be suitable for CO_2_ removal.

## 4. Conclusions

PEI/MEA impregnated FS adsorbents are promising candidates for separating CO_2_ from biogas. They are easily prepared from widely available and industrially produced materials and are able to adsorb CO_2_ efficiently. The PEI/MEA impregnated FS was found to exhibit much more promoting CO_2_ capacities than PEI impregnated FS adsorbents. These new adsorbents presented higher thermal stability than MEA impregnated FS adsorbents. Additionally, similar CO_2_ adsorption capacity was exhibited although different ratios of MEA to PEI were impregnated into FS, indicating that MEA could facilitate PEI to disperse evenly on the surface of FS and thus reduce the CO_2_ diffusion resistance. These FS-PEI/MEA adsorbents exhibited the attractive CO_2_ adsorption capacity of ca. 64.68 mg/g with a simulated biogas flow rate of 100 mL/min at 0.2 MPa and 25 ℃. An improvement of thermal stability would be achieved with the increasing value of PEI/MEA weight ratio. FS-15%PEI-5%MEA adsorbent could be regenerated at 130 °C, 0.2 MPa and an Ar (99.999%) flow rate of 200 mL/min for 15 min for at least 3 cycles without loss of CO_2_ adsorption activity. This approach of impregnating a mixture of MEA and PEI into porous supports may deliver new ideas on preparing amine impregnated porous materials, although further studies are necessary.

## Figures and Tables

**Figure 1 ijerph-17-01452-f001:**
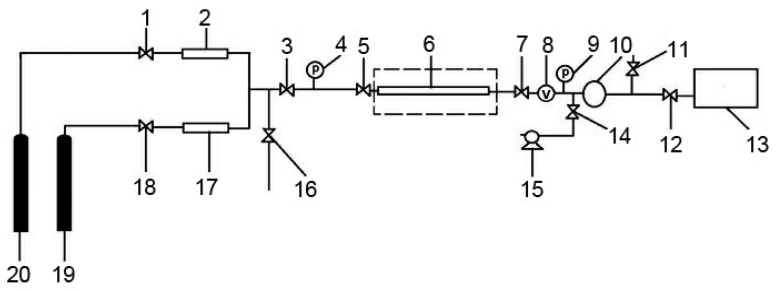
The schematic diagram of self-assembled fixed bed system. 1, 3, 5, 7, 11, 12, 14, 16, 18: cut-off valve; 2, 17: mass flow meter; 4, 9: pressure gauge; 6: adsorption column; 8: vacuum gauge; 10: back pressure regulator; 13: gas chromatography; 15: vacuum pump. 19: Ar gas cylinder; 20: CH_4_ and CO_2_ mixture cylinder.

**Figure 2 ijerph-17-01452-f002:**
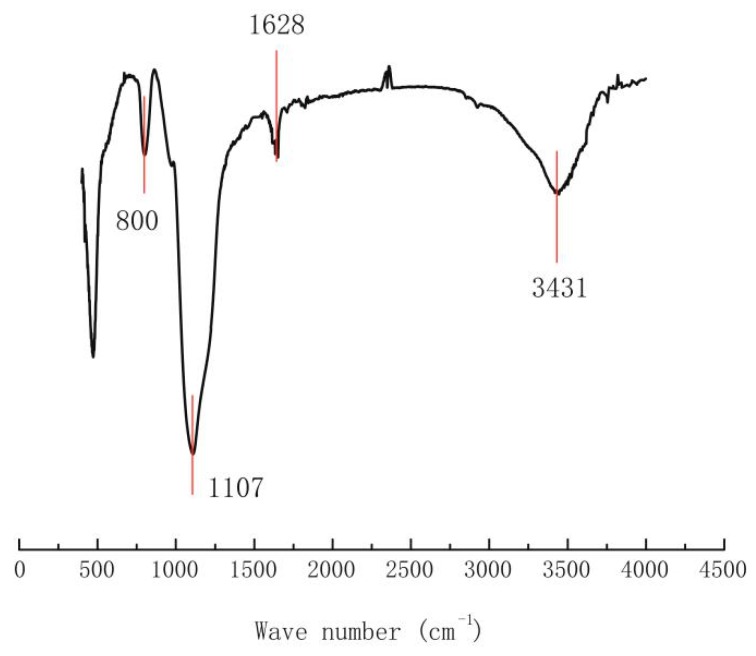
FTIR spectra of different silica.

**Figure 3 ijerph-17-01452-f003:**
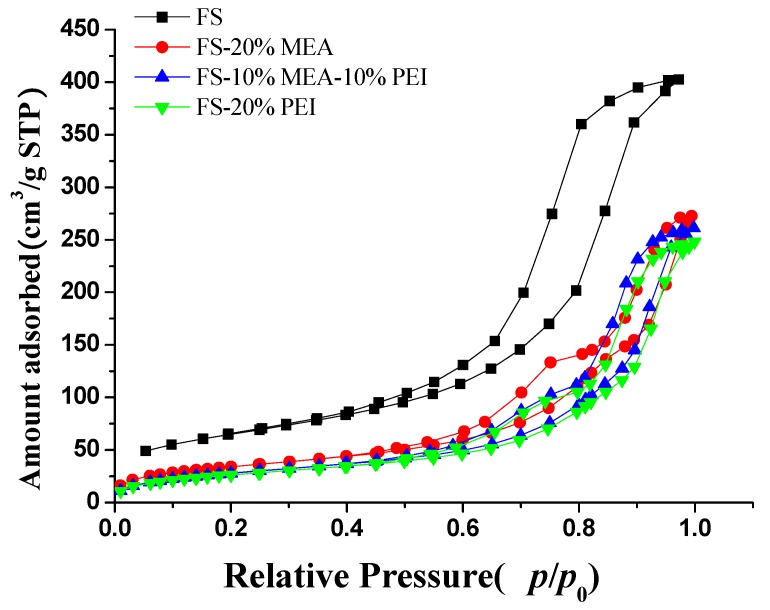
N_2_ adsorption–desorption isotherms.

**Figure 4 ijerph-17-01452-f004:**
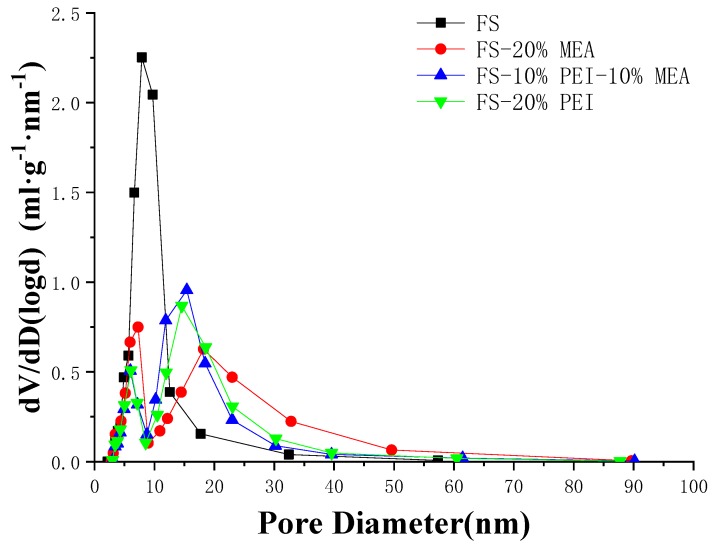
Pore size distribution.

**Figure 5 ijerph-17-01452-f005:**
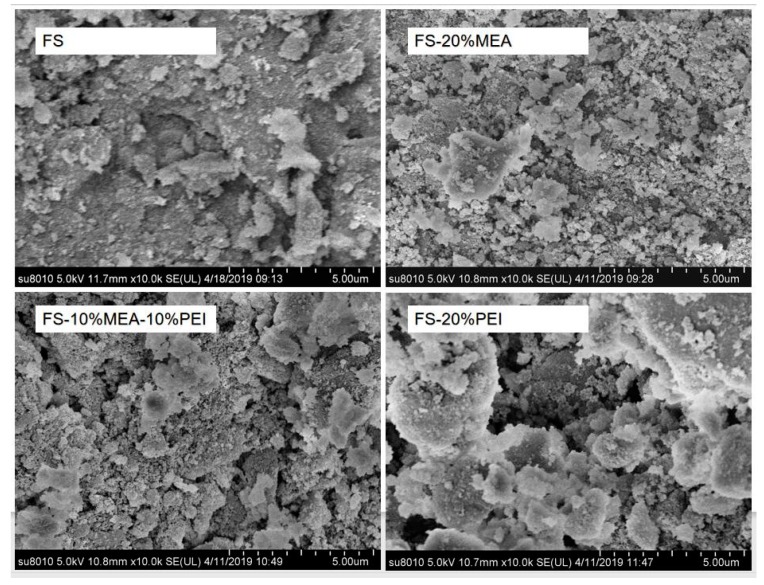
SEM images of different samples.

**Figure 6 ijerph-17-01452-f006:**
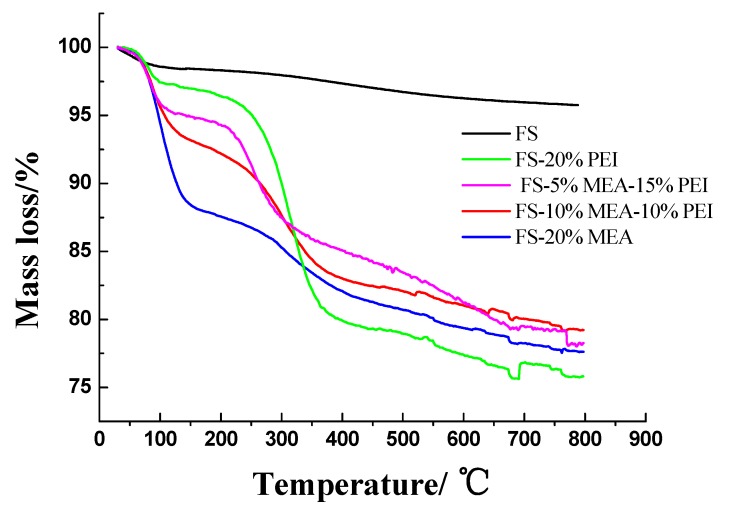
TG analysis of FNG-II silica (FS) before and after loading amine.

**Figure 7 ijerph-17-01452-f007:**
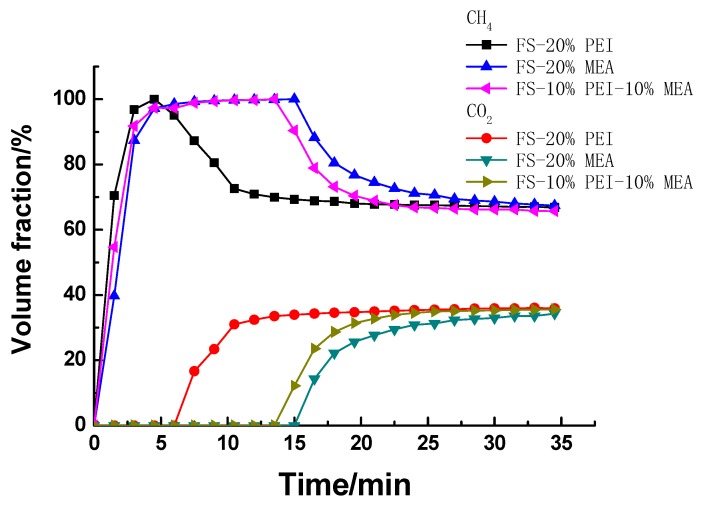
The breakthrough curves for CO_2_/CH_4._

**Figure 8 ijerph-17-01452-f008:**
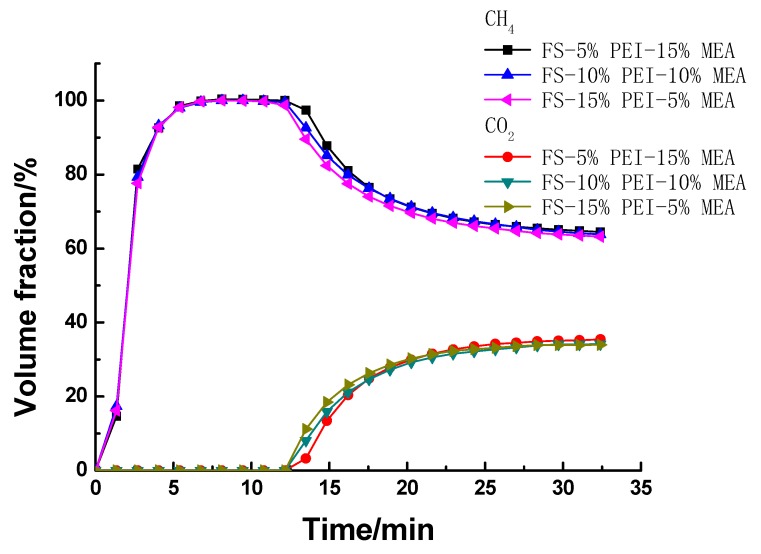
The breakthrough curves over FS functionalized by different ratios of polyethyleneimine (PEI) to ethanolamine (MEA).

**Figure 9 ijerph-17-01452-f009:**
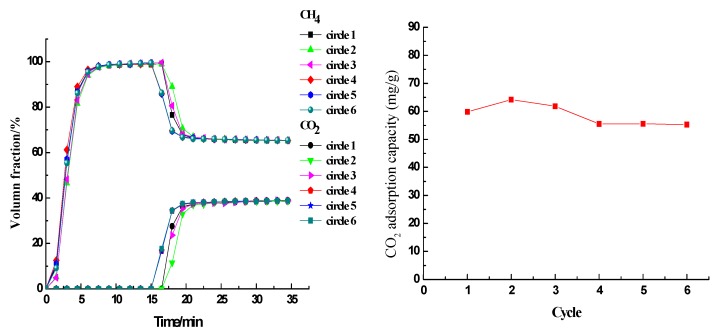
Adsorption/desorption cycles for FS-15%PEI-5%MEA.

**Table 1 ijerph-17-01452-t001:** Textural properties of the silica before and after impregnating organic amine.

Sample	BET Surface Area (m^2^·g^−1^)	Total Pore Volume (mL·g^−1^)	Average Pore Diameter (nm)
FS	230.2	0.6228	10.82
FS-20%PEI	97.6311	0.368724	15.10682
FS-20%MEA	121.8443	0.387443	12.71928
FS-10%MEA-10%PEI	103.8038	0.395548	15.24213

**Table 2 ijerph-17-01452-t002:** Comparison of adsorption capacities among various amine-modified adsorbents.

Adsorbent	CH_4_/CO_2_ (V/V%)	*T*^a^ (℃)	*p*^b^ (MPa)	*q*_ad_^c^ (mg/g)	Reference
FNG-II Silica-MEA(20%)	64.4/35.6	25	0.2	68.87	This work
FNG-II Silica-MEA(10%)-PEI(10%)	64.4/35.6	25	0.2	64.68	This work
FNG-II Silica-PEI(20%)	64.4/35.6	25	0.2	35.76	This work
MCM-41-PEI(50%)	-/100	75	-	111	[19]
SBA-15-PEI(50%)	-/100	75	-	127	[19]
D101-PEI(20%)	64.4/35.6	25	0.2	47.7	[33]
ADS-17-PEI(20%)	64.4/35.6	25	0.2	65.2	[33]
β-Zeolite-MEA(40%)	-/100	30	0.1	33.88	[37]
Activated carbon	50/50	25	0.12	64.68	[38]
amino-MIL-53(Al)	40/60	30	0.1	60.72	[39]

^a^*T* is the adsorption temperature (°C). ^b^
*P* is the total pressure (MPa) in the adsorption column. ^c^
*q*_ad_ is the CO_2_ adsorption capacity (mg/g) of the sample under specific conditions.

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
