# Peer review of "Enhanced Adsorption of Carbon Dioxide from Simulated Biogas on PEI/MEA-Functionalized Silica"

_ijerph, 2020, doi:10.3390/ijerph17041452_

Round 1

Reviewer 1 Report

Specific comments:

After reviewing the version of the draft, is possible to say that the document is well written, the experimental results and discussion sections are appropriately described, as well as it is consistent with the results analyzed.

Comment 1

EXperimental section

For clarity in the description of the separation tests it is recommended to include a scheme of the experimental setup used for the tests of CO2 adsorption.

Results and discussion.

Comment 2

It is suggested to include the structure of the amines used and identify in greater detail the wavelengths related to the main functional groups.

Comment 3

It is suggested to deepen the discussion on the decrease of the surface area and its effect on the selective adsorption of CO2.

Comment 4

The quality of the scanning electron microscopy image showed in figure 4 is poor and displays considerable surface charging effects, so it is necessary to change them.

Comment 5

For a better idea of the performance of materials in the separation of CO2 from the CO2 /CH4 mixture, it is suggested to present a comparative table with other materials used by the authors or materials reported in the literature and their discussion.

Reviewer 2 Report

In  this work, amino-based silica adsorbents were prepared, characterized and used to capture CO2. The manuscript is very interesting, however some questions should be addressed, as follow:

- Lines 13 and 19: What is FS and TDA???

- Line 44: “...from biogas is amine absorption.” It is correct? I suggest “...from biogas is adsorption on amino-funcionalized solids” .

- Introduction: Similar work has been recently published (reference 33). Compared to this report, do your research has some different or lightspot? How is this system different to other reports to merit publication? The authors should better explain if these amino-functionalized silica particles have been used in previous reports.

- Line 91: What is FNG-II and FS???

- Line 93: Ar is argonium?? Please, explain.

- Section 2.4: The authors should better explain the apparatus used in the continuous adsorption of CO2. Ratio mass/volume of fixed bed system, porosity, density, volumetric flow, Reynolds number, pressure drop across the reactor (Darcy’s law). In addition, a representative scheme of the fixed bed system is very important to the readers. The authors should explain why fixed experimental conditions have been used. Have been optimized in a previous report?

- In this work, amino agents were used at fixed concentration – 20% m/m. Why? The authors should better explain the experimental procedure to prepare these adsorbents.

- figure 1: FTIR should be changed – without interferences between 2000-1000 cm-1. It is hard to read it.

- A comparative study with previous reports is required to show the relevance of this study.

Reviewer 3 Report

The authors have synthesized a few novel sorbents, based on fumed silica as porous support and (combinations of-) MEA and PEI as active ingredient for the selective adsorption of CO2.

The idea of combining PEI with another, low molecular compound (as MEA or DEA or..) is not new, as also indicated by the authors. The results show a synergetic effect on the apparent CO2 capacity by using a combination of both ingredients. An (in-depth) analysis on "why" this is beneficial is unfortunately missing. 

The results as such are not very promising. For the targeted application sorbents with higher capacity (a factor 2-3 or more higher) do exist (see e.g. http://dx.doi.org/10.1016/j.seppur.2017.04.030). Also the number of cycles tested for stability (from the Conclusions: "...at least 3 cycles without loss of CO2 adsorption...") is too limited to have a claim for a stable sorbent.

Hence, in conclusion (i) the sorbent is not having a particular high capacity; (2) the stability is not proven (not clear...as only 3 cycles were tested); (3) the idea of combining different functional ingredients is not new.  So how to justify publication of this work ?!?

=>  In my view the authors could try to focus on studying and explaining how the interaction works and why, as they claim in line 208 ff, it was the PEI dispersion that caused the main synergetic effect. This could be true, but then the MEA is an unnecessary ingredient if the impregnation method is improved. 

Some detailed comments:

line 41/42:  "...delaying global warming..." :  depends strongly what you do with the CO2 you separate. If the delay is only the residence time in the separator it is not worth mentioning.

line 43:. "...Therefore, ...":   the preceeding arguments are no arguments that there is a need for a more economical and effective method. Comparison should be made against existing CO2 separating methods for biogas (such as membrane separation, water wash, amine scrubbing etc.). This is only partly done on p.2  

line 89-91:  First sentence of Materials section can be removed (not applicable)

section 2.4:  How did the authors check and validate their measurement technique ?   

line 159 (Table 1).:   Why is the pore diameter of the fresh FS smaller than that of the impregnated sorbents ????   This is counter-intuitive !!!

line 206-210:  Why do the authors claim that it is mainly the dispersion technique ?   Considering the amine-group density of both MEA and PEI, these are fairly similar and hence no big change on this aspect is to be expected!  On the other hand, if dispersion is the main thing....why then use MEA and not focus on improvement of the sorbent impregnation technique ?

Fig.8   Why are the capacities so modest?  the CO2 partial pressure (40% at 0.2 MPa) is fairly high and I would have expected higher capacities ?

line 235 (Conclusions):  in what sense is the sorbent 'a promising candidate'?  and to what extent does this lead to lower CO2 separation costs in comparison with earlier mentioned alternatives ?

Round 2

Reviewer 2 Report

The manuscript is finely done. Authors performed the reviewers suggestions and the manuscript is able to be published in IJERPH.